# Building Global Indigenous Media Networks: Envisioning Sustainable and Regenerative Futures around Indigenous Peoples' Meaningful Representation

**Reynaldo A. Morales [1,*], Dev Kumar Sunuwar [2,*] and Cristina Veran [3,*]**

1   Medill School of Journalism & Media, Buffett Institute for Global Affairs, Northwestern University, Evanston, IL 60208, USA
2   Kathmandu School of Law, University of Nepal, Biratnagar 56600, Nepal
3   Women's Media Center, New York, NY 10018, USA
*   Correspondence: ramoralesc@northwestern.edu (R.A.M.); dev.kumar@culturalsurvival.org (D.K.S.); verancm@gmail.com (C.V.)

**Abstract:** Asserting the right to meaningful representation, challenging the epistemological and methodological expansion of global corporate capitalism and its impacts on Indigenous Peoples' territories and cultures, aligns with the implementation of global treaties and conventions that are part of key international laws regarding issues of climate change, biodiversity conservation, education, global health, human rights, and sustainable development. Indigenous Peoples have been consistently excluded from nation state visions of modernity and development, which continues to limit their full participation in global sustainable development initiatives and their meaningful representation therein. Increasing the visibility of this struggle is imperative for Indigenous Peoples, particularly around the strategic areas in which the implementation of global sustainable development treaties, policies, and goals continues to affect their rights. This article inquires whether Indigenous Peoples' emancipatory appropriation of media means from a transnational perspective that breaks their regional enclosure can contribute to decolonize the world. More specifically, it questions how a new Indigenous global media network would contribute to decolonize the relations between Indigenous Peoples and nation states. A wider mapping of Indigeneity that decolonizes sustainable development becomes critical in order to formally document the efforts of Indigenous Peoples to reconstruct and restore their epistemic and material relations. This article questions how an Indigenous global media network around new nexus research can benefit Indigenous Peoples, and make visible the incorporation of the recommendations and principles from international law emanated from the self-determined voices of Indigenous leaders, experts, and policy makers to decolonize global sustainable development goals.

**Keywords:** Indigenous representation; Indigenous networks; nexus research; educational communication; decolonization studies; sustainable development; SDGs; Indigeneity; emancipatory media; UNDRIP; Indigenous Media Caucus

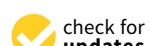



## 1. Decolonization and Governance Restoration

Beginning in the mid-20th century, mainstream media in settler colonial countries controlled by the dominant ethnic groups—which often presented and made reference to Indigenous Peoples as inferior, enemies of the state and of modernity and progress, and as representing a social problem—rose to become one of the most powerful social and cultural driving forces of the neocolonial dominion over Indigenous Peoples and territories, supporting the atomization and exploitation of their territories, creating untenable conditions for the surviving populations. Despite the incredible transformative impact of digital media and satellite communication technologies being used in the development of tribal and independent media through internet-based audiences in the new millennium, and the

advance of ethnic television which caters to specific interests, cultures, and even languages of immigrant communities in the global north, for the global mainstream masses, Indigenous Peoples remain condemned to oblivion and unable to speak to these audiences about their rights and struggles. This article focuses on the understanding of the opportunities for a decolonized media network to contribute to nexus research that addresses the global sustainable development goals. At the same time, it aims to promote the understanding of an academic understanding of the connections between transnational research around critical geography of Indigeneity, decolonization, education, and knowledge production, around movements and coalitions committed to transformative social change. These approaches include academic collaborations with Indigenous nations, and social movements, from a simultaneous critical interrogation of the colonizing forces that continue threatening Indigenous Peoples rights, seeing decolonization as a process of cultural and historical liberation (Hooks 1992).

As decolonization "once viewed as the formal process of handing over the instruments of government, is now recognized as a long-term process involving the bureaucratic, cultural, linguistic, and psychological divesting of colonial power" (Smith 1999, p. 98), building a strategic educational communication platform and investigative science journalism capacities represent vital mechanisms to monitor the relationship with nation states and non-Indigenous societies, to guarantee the survival and continuation of Indigenous Peoples' cultural, social and economic world. Embedding and interweaving anti-colonial or decolonization analysis into knowledge production and corresponding mobilization processes confronting neoliberalism remains a major challenge for organizing the actions against neoliberalism (Choudry 2007).

Global Indigenous media networks focused on their rights, territories, and their contributions to sustainable development are vital not only for Indigenous Peoples, but for governments, educational and research institutions, and multicultural societies in constant motion. Meaningful mainstream media representation of Indigenous Peoples is necessary to play a strategic role, assisting to include global Indigenous issues in the public agenda and facilitate UN agencies, governments, and NGOs, as well as educational and research institutions, to honor Indigenous Peoples' role in global sustainable development initiatives, as productive partners of environmental stewardship, cultural knowledge, and a biocultural relationship with the natural world.

The Indigenous Media and Communication Caucus, a thematic group of Indigenous journalists and other media practitioners, creators, and producers from around the world, was formed at the United Nations Permanent Forum on Indigenous Issues (UNPFII) in May of 2016, continuing an effort initiated in 2006, formalizing an initiative to address critical issues with regard to Indigenous Peoples and media representation. Muehlebach (2003) offers a critical interpretation of the processes and challenges by which Indigenous Peoples have consolidated their representation at UN and international law and policy forums under diverse perspectives:

> "Indigenous delegates at the UN have used the logic of decolonization to make the point that they represent colonized peoples who have fallen through the cracks of international law. While the "peoples" of the colonial regions of the world were granted independence, indigenous peoples, their representatives argue, remained "internal colonies" that were denied the inherent right to exist and to develop freely over time and according to their distinctive characteristics. They were denied the right to freely determine their relationship to the states they presently inhabit. This appeal makes up much of the moral power of indigenous interventions at the UN. However, it also represents a stumbling block in the process, for what looms large is the specter of the fragmentation of states and the dismemberment of their sovereign territorial integrity . . . equally ferociously being waged in international fora, in debates and negotiations surrounding its present and potential meaning. Scholars familiar with indigenous claims have argued that a real engagement with the aspirations of indigenous groups would

involve a much more complex actualization of self-determination than the law of decolonization and current dominant understandings of . . . the nature of the enduring relationships between self-determining indigenous communities and states". ([Muehlebach 2003](), pp. 247, 250)

The issue of meaningful representation emerges from these processes as an essential counterpart to the obligations that nation states have ratified through treaties and conventions, all of which have critical implications for the territorial and economic development rights of Indigenous Peoples. Since the 1980s, Indigenous Peoples in the Global South engaged in what was called "popular alternative communication" as a strategic platform through which to strengthen their languages, cultures, and identities, as well as to support their resistance against threats to their territories and ways of life, build new scenarios for the future, lead their own development projects, and also to establish new ways to interact with mainstream audiences. The Indigenous Media Caucus aims specifically to coordinate advocacy efforts to bring the challenges, triumphs, and achievements faced by Indigenous community media outlets, serving Indigenous communities and media practitioners, to the international stage and public agenda. Indigenous media initiatives around the world have demonstrated and emphasized the relevance of communication access, tools, and processes to link them to the assertion of their collective rights at multiple levels.

The United Nations Declaration on the Rights of Indigenous Peoples (UNDRIP) enacted in 2007 underscores, among its tenets on collective rights, the right to self-determination as a principle of the economic, social, and cultural development of world Indigenous Peoples. This right reflects and complements related rights as well, such as to participate in decision-making processes through their governance institutions on legislative and/or administrative decisions, as relevant to their needs and interests. Its implementation helps to affirm the mechanisms to free, prior, and informed consent for Indigenous Peoples over their own lands, territories, and resources, as well as regarding those rights that are social and cultural in scope, such as their traditions and cultural expressions.

Regarding media specifically, the Declaration's 16th Art. clearly establishes the unique and distinct right for Indigenous Peoples to have their own media (par. 1), and that privately owned media should also reflect cultural diversity among Indigenous Peoples (par. 2). Indigenous Peoples have the right to establish their own media in their own languages, and the equitable access to all other non-Indigenous media technologies, even though they may constitute a demographic minority; nation states shall provide support to ensure that national media coverage on sustainable development, climate change, biodiversity conservation, human rights, collective land rights issues, and other issues, duly reflects Indigenous cultural and social diversity without any form of censure. This article, which is part of a critical recommendation to UNPFII and other United Nation interagency networks, also establishes that nation states are bound, without prejudice, to ensure that Indigenous Peoples may exercise their full freedom of expression, encouraging also that such outlets are not editorially controlled by tribal governments or governance organizations, in order to adequately reflect Indigenous cultural and economic diversity.

Indigenous media as a social and technological phenomenon, despite the case of Native Hawaiian newspapers circulating for more than 150 years, has emerged in the last decade as a key contributor to monitor and ensure the implementation of sustainable development policies and practices, comprising a diversity of knowledge systems, beyond Western science, across the world. In the 1960s and 1970s, UN agencies such as FAO, UNESCO, and UNICEF, as well as international aid agencies, fostered the establishment of communication for development components in their projects, in order to help ensure that marginal populations are able to access information and education within the IEC (Information, Education, and Communication) model used, for example, in the sphere of public-health-related cooperation among many regions in Africa, Asia, and Latin America. As part of that process, the so-called "popular alternative communication" since the 1980s has provided a marginal forum reporting the participation of Indigenous Peoples in community training and capacity building venues, and remained a central community-

building mechanism within internal tribal and community-based networks created by different collectives, such as government, language and culture, spiritual and religious, historical societies and entertainment, etc. Some initiatives that started within public radio systems, labeled as "rural radio" and "radio for development", were conceived specifically for Indigenous populations.

"Are Indigenous Voices Being Heard? A Study on the State of Indigenous Community Broadcasting in 19 Countries" (Ramos 2019) was commissioned and reviewed by the Indigenous Media Caucus and published in 2020, jointly, by the internationally focused Boston-based NGO, Cultural Survival, and the Toronto-based NGO, World Association for Christian Communication (WACC.) Data used for the study, per individual country situation, were excerpted from yearbooks published by the International Work Group for Indigenous Affairs (IWGIA) from 2017 to 2019, and the report *"Tuning into Development: International Comparative Survey of Community Broadcasting Regulation"*, authored by Toby Mendel for UNESCO in 2013[1]. These sources included documents released by the International Seminar on Sustainability of Community Media: *Strengthening of Policies and Financing*, which was summoned by UNESCO and other organizations in Paris, France in 2015[2], and the *Report on the Situation of the Rights of Indigenous Peoples to Communication with Emphasis on Latin America, Black and Indigenous Communities*, prepared in 2019 by the Latin American Coordinator for Film and Communication of Indigenous Peoples (CLACPI), as well as academic publications and blogs produced by Indigenous organizations. The report concludes that,

> "Indigenous Peoples have been excluded from accessing media for many reasons, including their geographic location, languages, and legal barriers. Indigenous Peoples living in isolated areas have little physical access to urban-based centered media. Similarly, a lack of awareness of human rights, freedoms, and the rights to access information on state and municipal services further contributes to obstacles. Indigenous journalists work in difficult conditions in remote areas and are often the only mediums informing their communities on rights violations and cultural, environmental and social issues, which would otherwise be ignored by other media sources. Although all journalists face similar threats, it is often Indigenous journalists and communicators who are most impacted, as in most cases they work in informal settings in rural areas that are inaccessible to the mainstream media and even the government officials. They often lack access to protection mechanisms of justice. Indigenous journalists are generally not formally recognized as journalists because they do not have formal university training, or they are not affiliated with a major press or news agency. This additional safety risk often goes underreported and is overlooked by both government and international agencies." (Ramos 2019)

The study also highlights a common struggle faced by many non-profit Indigenous media organizations. Non-profit status and its reliance on donors, funders, and government assistance is not a universally accessible or widely sustainable practice—especially in large-population developing countries and those with many distinct Indigenous languages and nations. Accordingly, in order to pay for staffing (unless volunteer run), programming and operations costs, most media, especially newspapers and some radio and TV, must inevitably court advertising revenue to sustain itself. These experiences are different for non-profit, policy, and research organizations when related to international law and issues discussed at the United Nations level trying to influence regional and local policy changes, as well as human rights and environmental protections. Some non-profit Indigenous media initiatives, depending on context, depend on community volunteer contribution and engagement on every level, especially in their community-centered decision-making processes such as tribal referendums and consultations inherent to their governance systems and tribal, ethnic/land-based identities. This includes challenges to building and sustaining capacity in the management of human, technological, and fiscal resources, as well as in content programming. In the context of the Global North—in the US, Canada,

Australia, New Zealand, and Nordic countries—Indigenous media organizations have developed highly professionalized, educated and/or trained staff that earn some manner of professional salary that may be supported by their tribal governments or nation state programs.

For the most part, the emergence of Indigenous media has not depended on the power of technology and transmitters, their geographical coverage, the presence or absence of publicity, or whether they have access to the limited permits or licenses and the high costs imposed by private broadcasters, including digital television and rural radio channels available in their localities. Indigenous media has been instead focused on their self-determination right, serving their communities' need for reliable and timely information and, in the process, fostering collaborative relationships, community building, and identity affirmation with the limited resources at hand.

In this context, Indigenous Peoples' control of their own media representation must be understood as a key component of the restoration and/or assertion of self-determined governance, knowledge production and capacity building, and a mutual and equal acknowledgement among mainstream global audiences about the need for transitional forms of justice, and the important political, moral, and ethical imperatives in recognition of the survivors of colonization. "For its advocates, this would not be the socio-geographic separation of old, wherein indigenous peoples were robbed of an ability to pursue their definition of the good life. Rather, it would be a qualified form of place autarchy in which Indigenous Peoples achieve meaningful control over both the kind and the degree of interaction with non-indigenous peoples as needed" (Castree 2004, p. 158).

The issue of media representation, in terms of opportunities thereof, also relates to the implementation and enforcement of international free, prior, and informed consent (FPIC) standards to ensure "meaningful consultation", as established by ILO169, the Convention on Biological Diversity, UNDRIP, and multiple UN governance instruments for all relevant projects and activities. This is key in order to make policy design and decision-making processes more visible and transparent, so as to ensure the fullest participation of Indigenous Peoples and traditional customary authorities with emphasis on the plurality of cultural protocols. FPIC as a pragmatic principle outlines the most urgent and necessary conditions for sustainable development projects, and presents conflict-resolution mechanisms to be used to negotiate terms of consent (or rejection thereof) for extractive and infrastructure projects, as one example, so as to ensure the engagement of Indigenous governance systems and international law. For example, the restoration of biocultural rights in the management of natural and genetic resources with associated traditional knowledge is a crucial theme to be investigated and disseminated not among Indigenous communities in general, much less to communities of practice who are highly aware of these issues, but into mainstream audiences, mainstream press, specialized media in science and policy, national communication systems, academic journals, research projects, and United Nations communications system, so that they are adequately informed about the processes and issues inherent to the implementation of many collective rights such as FPIC. This includes the discussion of the differences between community self-consultation, private and corporate sponsored, nation state sponsored, and mixed state and corporate consultations regarding policies and projects that present potential or actual impact on protected areas, forests, waters, lands, and ecosystems where Indigenous Peoples live.

Indigenous Peoples are already extensively making use of global communications outlets such as internet-based news and social media applications to internally share news and information on a wide range of issues. However, these networks are still confined to internal tribal or community affiliation and membership, and to limited socialization to individualized consumption through laptops and tablets. Meanwhile, a wide fringe appropriation of media technologies by Indigenous media organizations and individual contributors and the vast content produced still remains invisible to larger audiences, and therefore Indigenous rights are still not considered in the mainstream public opinion. This is especially critical in order to challenge the persistence of neocolonial assertions that

continue to perpetuate the fallacy that Indigenous Peoples are gone, and their governance, knowledge, and cultural systems are the past.

To counterbalance the additional challenges created by the ultra-segmented media of the 21st century, it is urgently necessary to refine the scope of the content and media penetration outside of the typical internal and fringe communication networks created by Indigenous Peoples in commercial and non-profit media, in order to break the local representation silos and conquer presence in mainstream television around the key themes of their collective rights, their diverse contributions to global sustainable development, and the transparent and meaningful participation and consent of Indigenous Peoples in development programs and projects that affect their territories and governance.

## 2. National and Regional Indigenous Media Networks

Envisioning global media networks built around identities, alliances, fusions, and partnerships around Indigenous contributions to multiple areas of development, implies the material development of central repositories of multiple digital media experiences around online television (including apps for video and radio to use on smart devices and desktop computers) and multimedia blogs that combine diverse platforms and formats. This central content network would need to be centered around the articulation and exercise of the right of self-determination as expressed in international law, treaties, and conventions, from a strategic information, education, and communication (IEC) framework, promoting measurable changes in how the mainstream audiences as well as major press media understand and address Indigenous identities and cultural perspectives of governance.

The United Nations Indigenous Media Caucus seeks to develop avenues to assert the rights outlined in UNDRIP. One proposition of this envisioned global network is to generate awareness of and support for the implementation of free, prior, and informed consent processes to mitigate the negative impact of extractive industries such as mining and oil pipelines, or industries associated with roads, dams, mining, pipelines, and fracking infrastructure, among others, that threaten Indigenous territories, protected areas, and vital biodiversity. Indigenous media networks designed for mainstream television are also indispensable to change perceptions, attitudes, and actions in the fight against environment pollution, climate change, and damage due to large-scale agriculture and land grabbing, from the perspective of their governance systems. Media representation with emphasis on gender perspective and women's rights can also contribute to increase the visibility and possibly reduce violence against Indigenous women in rural areas calling for action and prevention. In addition, it can bring attention to everything from harmful cultural practices to urgent issues such as, for example, the visibility of human-trafficking operations targeting Indigenous women.

The report on Indigenous community radio referenced in this article devoted particular emphasis to issues of violence against women from the African continent, especially the crucial support of community radio stations managed by women's groups to promoting gender equity and women's human rights. Additionally, highlighted by this report are experiences documented from Guatemala, where a network of Mayan journalists developed programs to promote women's rights. A potential contribution of interconnected global Indigenous media networks is also to increase the visibility of peace-building processes around armed conflicts between Indigenous Peoples and states with comparable national contexts and legislations that lack protection of Indigenous Peoples, and even deny their existence in their jurisdictions, in violation of international law.

Three specific cases are cited in this paper to underscore the need for and contextualize the development of a new and globally engaged Indigenous media network in elevating issues relevant to Indigenous Peoples to the mainstream public agenda, in the nation states within which their territories exist, and counterbalance the absence of Indigenous programming and presence in mainstream media. Several examples, among the many developed by the diversity of Indigenous media organizations, are referred to here, in order to highlight a typical framework for collaborative participation and agreement among

multiple stakeholders, such that would not normally be possible in mainstream public and private media scenarios.

The first example is the Latin America Coordinator for Indigenous Peoples' Films and Communication (CLACPI) (Figure 1), integrated by many organizations in Latin America, promoting community-based media, particularly video production, as a way to preserve and enhance indigenous cultures from their own point of view. CLACPI was created in 1985 in Mexico by a group of ethnographic filmmakers with the idea of having a space to screen and discuss films about Indigenous Peoples. In their first festival, CLACPI members stated their goal of promoting Indigenous Peoples' appropriation of audiovisual media, addressing the "crisis of representation" that questioned the issue of "ethnographic authority" (Zamorano Villarreal 2017), towards an emancipatory vision (Schiwy 2009) that contributed to transforming political realities through training subaltern subjects in response to a regional and global tendency that reflected growing Indigenous movements and increasing agency, emerging from a decolonial cultural discourse and politics designed and promoted by Indigenous movements (Schiwy 2009), which in turn became a fertile ground to possibilities for academic research and political action (Zamorano Villarreal 2017).

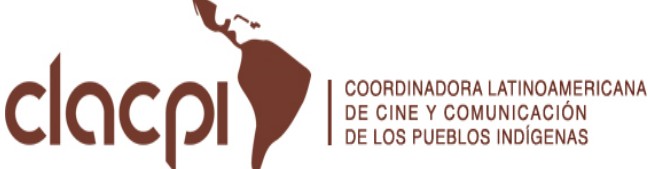

**Figure 1.** Indigenous Peoples Latin American Cinematography and Communication Coordinator. Source: www.clacpi.org, accessed on 11 September 2021.

CLACPI evolved to become a conglomerate of Indigenous-lead organizations from Bolivia, Chile, Argentina, Colombia, Cuba, Ecuador, Guatemala, Mexico, Peru, Venezuela, Nicaragua, and Brazil, and also from Catalunya and Romani Peoples, and Canadian First Nations. It also organizes the International Festival of Indigenous and First Peoples Cinematography and Communication (FICMAYAB) (Figure 2), which in 2017 hosted itinerant screenings of such works in schools, communities, and municipalities in Mexico, Guatemala, Honduras, Venezuela, Europe, and North America, expanding in 2018 to Panama to include the participation of the Abya Yala Indigenous Peoples from Panama and the Caribbean. Since 1985, CLACPI has organized Indigenous Film + Video Festivals about every two years. What makes this festival very special is that it is itinerant and international. Previous festivals include: México (1985 and 2006), Brazil (1987), Venezuela (1990), Peru (1992), Bolivia (1996 and 2008), Guatemala (1999), Wallmapu Chile (2004), Ecuador (2010), and Colombia (2012). The festival hosts a gathering of journalists and media producers to consult among one another on media actions and strategies. CLACPI represents a special case from Latin America that highlights the community-based aspect of Indigenous media representation promoted within living spaces for participation in the same communities.

The second case to note here is that of Isuma TV (Figure 3), a collective of inter-related Canadian Inuit-owned media entities based in Igloolik, Nunavut since 1990, with an office also in Montreal. Isuma started with an Inuit-focused media arts center, a youth media group, and a women's video collective. In 2004, they incorporated Isuma Distribution International, which in 2008 launched one of the first websites for Indigenous media arts, offering over 7000 films and videos in 84 languages. Maori TV in New Zealand had already launched an online television channel in 2004. By 2010, it launched *Digital Indigenous Democracy*, an online media network to inform and consult with Inuit people living in low-bandwidth communities facing imminent development.

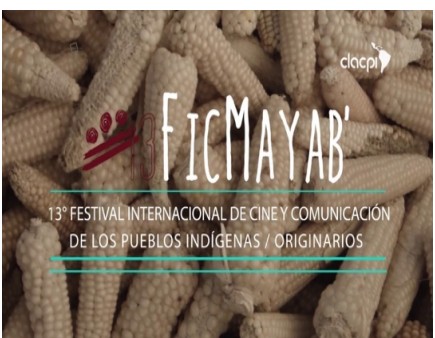

**Figure 2.** Indigenous Native Peoples International Festival of Cinematography and Communication. Source: www.clacpi.org, accessed on 11 September 2021.

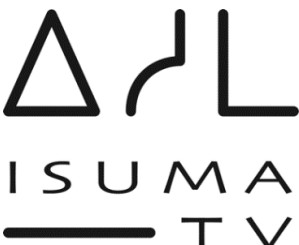

**Figure 3.** Isuma TV, a project of Isuma Distribution International Inc. Source: www.isuma.tv/isuma, accessed on 11 September 2021.

Recent Isuma projects include the feature drama *Maliglutit (Searchers)*, the TV series *Hunting with My Ancestors*, and the world's first Haida-language feature film, *SGaawaay K'uuna (Edge of the Knife)*. Their newest feature, *One Day in the Life of Noah Piugattuk*, was honored as a selection to represent Canada at the 2019 Venice Biennale, screened at the Toronto International Film Festival, and later awarded Best Canadian Film at the 2019 Vancouver International Film Festival. These experiences led the way for the creation of Uvagut TV, Canada's first Inuktut television channel, which currently broadcasts uninterrupted Inuit-produced programming from across Inuit Nunangat, including movies, news and information, cultural programs, current affairs, and archival media. The television channel airs nationally also on the Shaw Direct basic satellite channel 267, in Nunavut and NWT on Arctic Co-ops cable 240, in Nunavik von FCNQ cable 308, and online.

The third example presented here is Indigenous Television of Nepal (Figure 4), a community channel that reaches across this Himalayan country through fiber-optic cable networks, and the world beyond via a mobile app, website, and social media. Run by the Indigenous Media Foundation, a not-for-profit organization founded by Nepalese journalists belonging to the country's diverse Indigenous communities, the channel launched on the occasion of the World Indigenous Peoples Day on the 9th of August 2016, as a 24 h source of news, views, educational content, entertainment, and other informative programs on issues of interest and importance, created for, by, and of, and in the languages representing Nepal's Indigenous Peoples (known there as Adivasi Janajati). The Nepalese government classifies these into 59 distinct ethnic groups, each being the custodian of its distinct culture, language, spiritual practices, traditional knowledge, and biological diversity.

Indigenous Television of Nepal is a unique case within South Asia, in the manner in which it provides representation for communities in their respective mother tongues, as well as being grounded in their traditional cultures and written and oral histories, while also recognizing and including the traditional homeland and geographical cluster of each. Adivasi Janajati comprise 36 percent of the country's total population of 26.8 million. Additionally, at least 25 more ethnic communities are engaged in the process of being recognized as Indigenous Peoples. Of the 123 different languages spoken by 125 ethnic and caste groups in Nepal, 95 percent are spoken by Indigenous Peoples. From its inception,

Indigenous Television of Nepal has produced and broadcast TV shows, news, music videos, feature films, and documentaries in fourteen of these Indigenous languages (Bantawa-Rai, Chamling-Rai, Kulung Rai, Tamang, Sunuwar, Magar, Newar, Limbu, Sherpa, Hyolmo, Thami, Chhantyal, Yakkha, Gurung), with the goal to educate and connect the country's Indigenous Peoples with one another, facilitating dialogue and promoting mutual understanding among Nepal's diverse ethnic groups.

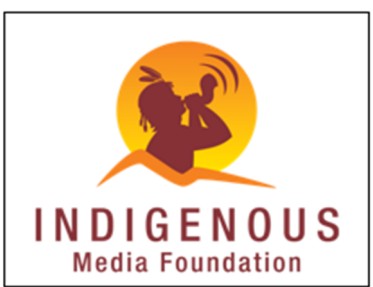

**Figure 4.** Indigenous Media Foundation, a not-for-profit organization of Nepal that supports Indigenous Television of Nepal. Source: www.indigenousmediafoundation.org, accessed on 11 September 2021.

## 3. Envisioning a Policy-Based Global Indigenous Mainstream Television Network

The meaningful cultural, social, economic, and political representation of Indigenous Peoples in all matters that impact them requires their access to and voice in mainstream media, beyond their existing Indigenous spaces, as part of external audience development. Conquering these new spaces is key to educating non-Indigenous audiences, the general public, governments, and also institutions, about issues around the pending stages of UN-DRIP implementation, as well as that of United Nations mandates and recommendations (Figure 5) related to Indigenous Peoples rights that affect them, as a result of colonization, genocide, material dispossession, and displacement.

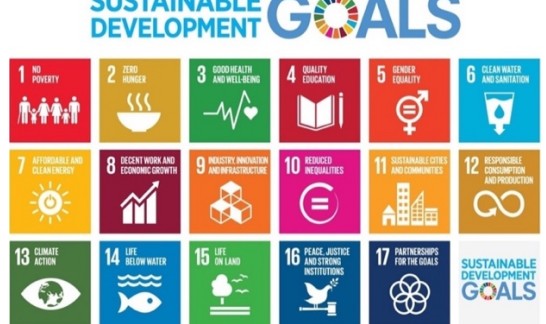

**Figure 5.** The 17 Sustainable Development Goals identified by science research and policy panels gathered around the United Nations interagency system as the global priority. Source: www.commons.wikimedia.org/wiki/File:Sustainable_Development_Goals.png, accessed on 11 September 2021.

A central Indigenous network needs also to provide real-time coverage of Indigenous Peoples' press agencies around the world, based on the retransmission and reformatting of content provided by Indigenous press rooms, including coverage of actions and outcomes around sustainable development, human rights, land rights, environmental issues, global health, gender equality, nutrition, and education, all in regard to the priorities of the 17 global sustainable development goals, including the discussion of dominant and subjugated knowledge and its representation, bringing to light that knowledge that has been suppressed and disqualified in knowledge hierarchies (Foucault 2020), has been made invisible. Research informed by postcolonial studies, critical educational research, and sustainable development, among other related disciplines, has demonstrated that Indigenous Knowledge Systems (IKS) are critical to sustainability and also to the survival

and thriving of Indigenous Peoples and Local Communities (IPLCs), as many of their knowledge systems are approached in research, industry, and policy in oppressive and extractive ways.

This discussion connected to the condemnation to the expansion of global corporate capitalism and its impacts on Indigenous Peoples territories and cultures has intensified in recent years, owing in part to the increasing emphasis on the biocultural research methods and sustainable development frameworks that are perceived to be vital for global biodiversity conservation and in response to climate change. The biodiversity framework presents correlations between biodiversity loss to issues of health, nutrition, community development, economic development, and gender equity, as the "loss of biodiversity is likely to disproportionately impact on the health and well-being of the poorest" (Daw et al. 2011). Therefore, a wider mapping of Indigeneity that connects with sustainable development is critical in order to formally document the efforts of Indigenous Peoples to reconstruct and restore their knowledge systems and negotiate forms of coexistence with non-Indigenous societies and nation states.

The economic dimension of the exclusion of Indigenous Peoples in globalized industrial development also needs to be addressed in the public agenda, as alternative models and measures of economic welfare, such as inclusive wealth accounting, natural capital accounting and degrowth models, are increasingly considered as possible approaches to balancing economic growth and the conservation of nature and its contributions and to recognizing trade-offs, the pluralism of values, and long-term goals (Bongaarts 2019). Degrowth is defined as a reduction in production and consumption at the local and global levels, and its main assumption is that human progress without economic growth is possible (Schneider et al. 2010). Economic degrowth, far from being an option, is an inexorable trend, since, due to the depletion of natural resources and environmental destabilization of the conditions that prevailed in the Holocene, global economic growth rates have already begun to decline (Marques 2016). This framework is deeply related to participatory biodiversity conservation, as it considers local people as key decision makers in conservation planning and in the access and benefit sharing of genetic resources derived from traditional knowledge.

A new breakthrough report (Wright 2020) estimated that not only may "the use of traditional knowledge increases the efficiency of screening plant resources for medicinal purposes by more than 400 per cent", but also "[the value] of the world market for medicinal products derived from leads associated with traditional knowledge is estimated at approximately USD 43 billion".[3] Furthermore, it estimates that "benefits of approximately USD 5.4 billion would flow to Indigenous and local communities around the world if multinational corporations paid royalties for traditional knowledge used in food, agriculture and pharmaceuticals."[4] These realities are virtually unknown to the general public, however, due to the absence of media attention and analysis of their impact on the life and survival of Indigenous Peoples today.

The envisioned strategy of a global media network would also include the retransmission and design of sub-channels and libraries, and of curated television and radio content from prioritized sources such as Indigenous stakeholder groups. The specific policy-based connection of Indigenous media initiatives to both ratified conventions and national goals requires support from national, regional, and supranational bodies as important components for mainstreaming the actions and agreements between nation states and Indigenous Peoples, for the fulfillment of global sustainable development goals, with the full inclusion and support for Indigenous Peoples' participation therein. The organizational structure of a such a global media network would require its division into autonomous but complementary and coordinated units, based on content provided by Indigenous media practitioners through a central media server and curated into subchannels. Each media system would need to create a subdivision of organized content in areas such as international news, education, culture, traditional knowledge, language, health, sustainable agriculture, nutrition,

economic development, and agroecology, among others, relevant to the self-determined representation of Indigenous Peoples governance.

OTT (Over the Top) platforms, connected through a centralized management server for user registration and account management, can securely connect contributors from multiple regional participant networks. Each management server would support sub-channels divided by geographical regions, with front ends on multiple connected devices, such as smart phones and streaming television. Elevating Indigenous media visibility in OTT platforms will thereby allow it to bypass cable, broadcast, and satellite television platforms—entities that have traditionally acted as gatekeepers to control the distribution of content, unconcerned with, if not outright oppositional to Indigenous representation therein.

This endeavor requires legitimate continental and regional representation of Indigenous Peoples of the world formally congregated around UNPFII and affiliated forums and conventions, with special appeal to the United Nations interagency community and Indigenous governments and organizations that have sustained a relation with nation states and policy making for the last three decades, as a result of these collective forums. The Indigenous Media Caucus seeks the aforementioned support from nation states that ratified these conventions to ensure that each continent and region can develop and strengthen the human and technological capacity to coordinate the collection and curation of media content, to design a programming structure according to the needs of Indigenous Peoples representation, and ultimately work in consultation with key organizations such as CLACPI, Isuma TV, and Indigenous Television of Nepal, and others, as well as UN agencies, NGOs, and inter-governmental sources.

A Global Indigenous Media Network composed of Indigenous media organizations formally gathered around UNPFII, funded by direct contributions from more than 140 nation state signatories of UNDRIP, is critically needed to disseminate content outside of Indigenous internal cultural and social networks and into global mainstream media networks, to forward into public agenda new pressuring issues recognized by international law since the first "Study of the Problem of Discrimination Against Indigenous Populations: Final Report", which was introduced to the UN Economic and Social Council Commission of Human Rights by the UN Special Rapporteur, Mr. José Martínez Cobo in 1981 (Cobo 1986).

The potential for Indigenous Peoples participating in a network of networks, to transform the media representational paradigms, shows new routes and meaning from localized media experiences into central global OTT streamline television channels, expanded to multiple digital cable formats, offering an articulated retransmission of archive and current content already produced exclusively by Indigenous Peoples with a unified goal around their collective rights. This centralized global Indigenous television network would offer opportunities for Indigenous media organizations to break the historical regional isolation and local marginalization of Indigenous Peoples from around the world, working with human rights institutions on guidelines for the implementation of the United Nations Declaration on the Rights of Indigenous Peoples, and to best ensure their access to and representation within the media overall. In turn, this can help to ensure the discussion of their rights is prioritized in key educational, social, and cultural institutions, networks, and governance systems. Furthermore, it can lead to the development of an international press platform from which Indigenous organizations—especially populations in isolated and remote areas—can launch their own projects, events, and reports.

This level of interrelation would require a global communication strategy backed by UN agencies, as the action of Indigenous media networks should be legally bound to treaties such as ILO 169, in which a delegate to the International Labor Conference ILC, or the Governing Body, can file complaints against a State for non-compliance with a convention it has ratified; the United Nations High Commission for Human Rights (OHCHR), UNESCO, United Nations Environmental Program UNEP, and the Convention on Biological Diversity, with current open negotiations between nation states and Indigenous Peoples'

representatives in regard to the protection of ecosystems and natural resources in national policies, the protection of traditional knowledge, and, in particular, the access and benefit sharing agreements around genetic resources with immense implications for Indigenous Peoples' economic development and regenerative futures.

**Author Contributions:** Conceptualization, decolonization theory and international policy around Indigenous meaningful representation applied to global Indigenous media representation; methodology, qualitative analysis and contextual discourse analysis; software, Mendeley; validation, the experience compiled in this article is part of participation at the United Nations Forum on Indigenous Nations and the UN Indigenous Media Caucus; formal analysis, the textual and visual references in this article were applied to organize the contributions from international media experiences connected to the UN Indigenous Media Caucus; investigation, these reflections are the result of envisioning policy changes in the application of the principle of meaningful representation of Indigenous Peoples reaffirmed by international law and policy reflected in the current status of mainstream and alternative media and communication; resources; data curation, peer review process; writing—original draft preparation, led by R.A.M.; writing—review and editing, R.A.M., D.K.S. and C.V.; supervision, R.A.M.; project administration, R.A.M. All authors have read and agreed to the published version of the manuscript.

**Funding:** This research received no external funding.

**Institutional Review Board Statement:** Not applicable.

**Informed Consent Statement:** Not applicable.

**Conflicts of Interest:** The authors declare no conflict of interest.

## Notes

1    Mendel (2013).
2    (UNESCO 2015).
3    Posey and Plenderleith (2002).
4    United Nations Development Program and Rural Advancement Foundation International (1994).

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
