# Peer review of "Building Global Indigenous Media Networks: Envisioning Sustainable and Regenerative Futures around Indigenous Peoples’ Meaningful Representation"

_humanities, doi:10.3390/h10030104_

Round 1
Reviewer 1 Report
The topic and the title are very relevant, but the piece is very descriptive (almost like a journalistic report) and it is difficult to find the theoretical framework and the specific analysis. It would have been interesting to explore in some detail some of the 3 examples, especially the first one that has to do with Latin America (to be more align with the special section) and explore some of the visual cultural manifestations emerging from CLACPI.
Some fragments are problematic from a decolonial and ecocritical perspective. For example, in line 346, the essay is not critical with the concept of sustainable development. This concept has been widely criticized by Latin American decolonial and indigenous scholars (see postdevelopment critique).
Similarly, in lines 374-385, the essay embraces the concept of economic growth as something unquestionably desirable and not as the root problem. Again, a degrowth and decolonial critique is lacking here.
Author Response
Point 1. Some fragments are problematic from a decolonial and ecocritical perspective. For example, in line 346, the essay is not critical with the concept of sustainable development. This concept has been widely criticized by Latin American decolonial and indigenous scholars (see postdevelopment critique).
Response: I have incorporated more clear decolonial and ecocritical perspectives. I corrected the approach in line 346 to specifically address the need to decolonize sustainable development and the sustainable development goals. I also included a postdevelopment perspecive aligned with this approach.
Point 2. Similarly, in lines 374-385, the essay embraces the concept of economic growth as something unquestionably desirable and not as the root problem. Again, a degrowth and decolonial critique is lacking here.
Response: I have corrected this confusion to include the perspective of degrowth and decolonial (lines 432-446) critique to specifically make reference to to the expansion of global corporate capitalism and its impacts on Indigenous Peoples territories and cultures. I included a definition of economic degrowth to make clear this connection.
Point 3: I have expanded the information about CLACPI as a prominent example from Latin America, highlighting its contribution to a regional consolidation of experiences linked to the emergence of Indigenous movements, as well as decolonial cultural discourse and politics designed and promoted by Indigenous movements.
The article cannot focus on the particular Latin American region as it needs to establish the critical urgency of a global media platform that forges ahead the already consolidated global Indigenous representation at UN forums and conventions to address a global response to common issues and struggles. The UN Indigenous Media Caucus has emerged from this global coalition from the recognition that no experience is more important than the the other and that we face a common front of action that requires a unified perspective.
Reviewer 2 Report
Article: ‘Building Indigenous Media Networks: Envisioning Sustainable and Regenerative Futures around Indigenous Peoples’ Meaningful Representation’
This noteworthy and valuable piece on indigenous media networks advocates comprehensively and convincingly for the creation of global networks to reach beyond indigenous networks and to raise awareness of the issues that are fully relevant to the indigenous peoples in the nation states within which they inhabit. In this sense, the article provides the reader with a clear and thorough overview of some of the key players, of the main challenges indigenous media networks face, and of potential answers and strategies that can be adopted. The author focuses on the work carried out by the network Indigenous Media Caucus with particular focus on the report ‘Are Indigenous Voices Being Heard? A Study on the State of Indigenous Community Broadcasting in 19 Countries’ (pp. 3-5) and covers three specific examples (pp. 6-8) to underline ‘a typical framework for collaborative participation and agreement among multiple stakeholders’ (p. 6).
The paper has potential to be published, but for this reader it will only reach publishable quality if there are some substantial revisions to the current version. First, many sections read like a ‘white paper’, that is, an in-depth report informing readers about the topic under considerations/ Other sections and paragraphs read like recommendations that one can find in a policy paper. Second, and in relation to the tone and approach of the piece, the author does not establish in an explicit manner what the research issues or questions s/he intends to address, the contribution the article will make to work in this field or how they relate to existing work. And, third, one of the most problematic aspects of the piece is working out who the reader of this piece is. I assume that they are likely to be scholars in the field of Iberian and Latin American Cinemas and Television (as per the Call for Papers) or scholars with an interest in the broader field of Indigenous Peoples (as per the broader points made throughout the paper). When rewriting and finessing the piece, it might be important to consider what these scholars have to learn from the article.
Following on from the above, I recommend the following revisions:
- a revised abstract which provides a clear summary of the article’s purpose (research questions, central argument and case study/case studies). This should be accompanied by a reduction in the number of keywords.
- a focused introduction which sets up aims, methodology, object of study, the contribution the article will make to work in this field and its relation to existing work. In other words, an introduction that responds to the expectations of a scholarly article. For example, could the paper be framed around the three cases discussed in pp. 5-8 and therefore foregrounded in the introduction? Or, in order to fit with the Call for Papers, could a Latin American dimension work as a running thread (looking at the webpage of the Indigenous Media Caucus among the founders and existing members of the Indigenous Media Caucus there are representatives of Latin America (Guatemala, Venezuela and Peru)).
- further use of subheadings / intertitles to break the dense prose and to take the reader by the hand, in particular through the first five pages.
- a much stronger conclusion which presents research findings rather than recommendations.
Attention to these will inevitably require a major rewrite of the article, which is why I have preferred this option over minor revisions. But, overall, I do think there is the kernel of an interesting and worthy article here.
Author Response
Thank you very much for your detailed review.
Point 1. A revised abstract which provides a clear summary of the article’s purpose (research questions, central argument and case study/case studies). This should be accompanied by a reduction in the number of keywords.
I have corrected and redrafted the abstract to provide an upfront in-depth report informing readers about the topic under considerations. The paper is intertwined with the emergent and current work around policy. The action of this proposed network is directly related to that work as it is part of the emergent coalition of the UN Indigenous Media Caucus.
Point 2. One of the most problematic aspects of the piece is working out who the reader of this piece is. I assume that they are likely to be scholars in the field of Iberian and Latin American Cinemas and Television (as per the Call for Papers) or scholars with an interest in the broader field of Indigenous Peoples (as per the broader points made throughout the paper). When rewriting and finessing the piece, it might be important to consider what these scholars have to learn from the article.
Response: The article responds to a real struggle from all regional media experiences and initiatives in the last three decades to transcend their regional enclosures and reach a global impact that honors the urgency expressed by the emergent coalition of the international Indigenous representation in international law forums and conventions in the face of their common struggles and opportunities. Even that the readers of this journal would be predominantly from Latin American Studies, the explicit connection made here is vital as the transnational perspective (rather than comparative) represents an urgent approach that needs to be understood more specifically around world Indigenous Peoples rights and decolonization. I have made a conscious decision to avoid a miscellaneous focus on a regional Latin American focus to highlight a transnational perspective that honors the current work that Indigenous Peoples are carrying. The reality is that all are learning and inspiring each other and their interrelation is not only a necessity but a reality.
The article has expanded the background about CLACPI to make more clear references about the important of this lead initiative in establishing a continental coalition that responded to the emergent Indigenous movements and the decolonizing efforts in the region to build Indigenous capacities towards the media means appropriation across the world.
Point 4. A focused introduction which sets up aims, methodology, object of study, the contribution the article will make to work in this field and its relation to existing work. In other words, an introduction that responds to the expectations of a scholarly article. For example, could the paper be framed around the three cases discussed in pp. 5-8 and therefore foregrounded in the introduction? Or, in order to fit with the Call for Papers, could a Latin American dimension work as a running thread (looking at the webpage of the Indigenous Media Caucus among the founders and existing members of the Indigenous Media Caucus there are representatives of Latin America (Guatemala, Venezuela and Peru)).
Response. The article made a corrected introduction that sets up aims, methodology, object of study, the contribution the article will make to work in this field and its relation to existing work. At the same time, the article intends to establish a transnational perspective around the intersection between Indigenous international law and policy and the need for a unified media representation linked specifically to what is expressed in these realms. The Indigenous representation at UN forums and conventions and their emergent relation to a global decolonial action and mobilization is the theme of this article. My work as new Northwestern University Faculty Fellow of the Buffett Institute for Global Affairs, and my work at Indigenous delegate at the Convention of Biological Diversity, Climate Change Convention, Human Rights and High Level Political Forum for Sustainable Development is connected specifically to establish the interconnections of this work to forge ahead the urgently needed global representation in support of the work around policy making currently in place in simultaneous fronts.
Point 5. Further use of subheadings / intertitles to break the dense prose and to take the reader by the hand, in particular through the first five pages.
Response. I worked on making more clear transitions between different sections and paragraphs. Please let me know if you still want me to use clear subheadings of intertitles.
Point 6. A much stronger conclusion which presents research findings rather than recommendations.
Response. I have corrected the conclusions to more explicitly present the findings resulted from our work around nexus research, policy, media representation and to highlight the critical impact that this initiative fosters.
Round 2
Reviewer 2 Report
First of all, I would like to thank the author for providing a clear and sound response to my proposed revisions. It makes the second reading much more focused and facilitates the reviewer’s identification of those sections where significant revision has been undertaken.
This is a much-improved version. The author highlights both in the abstract and in the body of the argument (introduction, case study and conclusions) how this scholarly intervention sits vis-à-vis current work on policy and how it contributes to a wider mapping of Indigeneity and to decolonising efforts and agendas. The theoretical and methodological importance of a transnational approach responds, according to the author, to the need to adopt ‘an urgent approach that needs to be understood more specifically around world Indigenous Peoples rights and decolonization’ rather than a comparison taking case studies drawn from the Latin American context as the main focus. Perhaps this is something that the editor(s) of the Special Issue may have to point out in their Introduction so that they frame appropriately and convincingly the function of this piece in the collection of articles.